# Tissue-Specific Microparticles Improve Organoid Microenvironment for Efficient Maturation of Pluripotent Stem-Cell-Derived Hepatocytes

**DOI:** 10.3390/cells10061274

**Published:** 2021-05-21

**Authors:** Ensieh Zahmatkesh, Mohammad Hossein Ghanian, Ibrahim Zarkesh, Zahra Farzaneh, Majid Halvaei, Zahra Heydari, Farideh Moeinvaziri, Amnah Othman, Marc Ruoß, Abbas Piryaei, Roberto Gramignoli, Saeed Yakhkeshi, Andreas Nüssler, Mustapha Najimi, Hossein Baharvand, Massoud Vosough

**Affiliations:** 1Department of Stem Cells and Developmental Biology, Cell Science Research Center, Royan Institute for Stem Cell Biology and Technology, ACECR, Tehran 1665659911, Iran; ensieh_zahmatkesh@yahoo.com (E.Z.); zahrafarzaneh2006@yahoo.com (Z.F.); zahrabiology85@gmail.com (Z.H.); f.moeinvaziri@yahoo.com (F.M.); s.yakhkeshi@modares.ac.ir (S.Y.); 2Department of Developmental Biology, University of Science and Culture, Tehran 1665659911, Iran; 3Department of Cell Engineering, Cell Science Research Center, Royan Institute for Stem Cell Biology and Technology, ACECR, Tehran 1665659911, Iran; biomaterialist@gmail.com (M.H.G.); ibrahimz_edl@yahoo.com (I.Z.); mj.halvaei@yahoo.com (M.H.); 4Department of Traumatology, Siegfried Weller Institute, University of Tübingen, 72076 Tübingen, Germany; othman.amnah@web.de (A.O.); m.ruoss@hotmail.de (M.R.); andreas.nuessler@googlemail.com (A.N.); 5Department of Tissue Engineering and Applied Cell Sciences, School of Advanced Technologies in Medicine, Shahid Beheshti University of Medical Sciences, Tehran 1985717443, Iran; piryae_a@yahoo.com; 6Department of Biology and Anatomical Sciences, School of Medicine, Shahid Beheshti University of Medical Sciences, Tehran 1985717443, Iran; 7Division of Pathology, Department of Laboratory Medicine, Karolinska Institutet, 17177 Stockholm, Sweden; roberto.gramignoli@ki.se; 8Laboratory of Pediatric Hepatology and Cell Therapy, Institute of Experimental & Clinical Research, Université Catholique de Louvain, B-1200 Brussels, Belgium; 9Department of Regenerative Medicine, Cell Science Research Center, Royan Institute for Stem Cell Biology and Technology, ACECR, Tehran 1665659911, Iran

**Keywords:** liver organoid, tissue-specific microparticle, pluripotent stem cell, hepatic differentiation, tissue engineering

## Abstract

Liver organoids (LOs) are receiving considerable attention for their potential use in drug screening, disease modeling, and transplantable constructs. Hepatocytes, as the key component of LOs, are isolated from the liver or differentiated from pluripotent stem cells (PSCs). PSC-derived hepatocytes are preferable because of their availability and scalability. However, efficient maturation of the PSC-derived hepatocytes towards functional units in LOs remains a challenging subject. The incorporation of cell-sized microparticles (MPs) derived from liver extracellular matrix (ECM), could provide an enriched tissue-specific microenvironment for further maturation of hepatocytes inside the LOs. In the present study, the MPs were fabricated by chemical cross-linking of a water-in-oil dispersion of digested decellularized sheep liver. These MPs were mixed with human PSC-derived hepatic endoderm, human umbilical vein endothelial cells, and mesenchymal stromal cells to produce homogenous bioengineered LOs (BLOs). This approach led to the improvement of hepatocyte-like cells in terms of gene expression and function, CYP activities, albumin secretion, and metabolism of xenobiotics. The intraperitoneal transplantation of BLOs in an acute liver injury mouse model led to an enhancement in survival rate. Furthermore, efficient hepatic maturation was demonstrated after ex ovo transplantation. In conclusion, the incorporation of cell-sized tissue-specific MPs in BLOs improved the maturation of human PSC-derived hepatocyte-like cells compared to LOs. This approach provides a versatile strategy to produce functional organoids from different tissues and offers a novel tool for biomedical applications.

## 1. Introduction

End-stage liver diseases account for almost two million deaths per year worldwide [1]. Moreover, drug-induced liver injury, as a public health concern, remains a potential health challenge [2]. It was reported that between 1953 and 2013, over 18% of pharmaceutical withdrawals occurred due to hepatotoxicity [3]. Animal models are used for drug screening, discovery, and toxicity testing, and they offer numerous advantages; however, they are often limited in relevance, time-consuming, and expensive and raise concerns [4,5]. Alternatively, in vitro culture of tissue-specific cells has attracted considerable attention as a promising approach to develop biomimetic systems for the prediction of potential hepatoxicity of drugs [6].

Traditional two-dimensional (2D) monolayer cells, cultured on flat and rigid substrates, are usually used for cell-based assays in drug development. Since almost all cells in the in vivo environment are surrounded by other stromal cells and ECM in a 3D condition, 2D cell culture does not adequately recapitulate the natural 3D environment of cells. In addition, it should be considered that physiological cell–cell interactions that typically happen in a 3D culture influence cell signaling, polarity, viability, and drug response [7,8]. Recently, liver organoids (LOs) were introduced as promising in vitro models for drug discovery, toxicology studies, disease modeling, and cell-based therapy treatments for patients suffering from hepatic failure [9,10]. The main principle of LOs is the recapitulation of major components and developmental steps of organogenesis or tissue repair in a dish [11]. In a pioneer work, Takebe and colleagues showed that the coculture of PSC-derived hepatic endoderm (HE) with mesenchymal stem cells (MSCs) and human umbilical vein endothelial cells (HUVECs) on a 3D Matrigel matrix led to condensation into a multicellular aggregate with liver-mimicking cell organization, called a liver bud organoid [12]. This strategy to develop a liver organoid was not scalable or reproducible to produce cells in clinically relevant numbers. Moreover, homogenous and efficient diffusion of nutrients, wastes, and other soluble factors throughout the aggregates might be restricted due to their large size [8]. Hence, a large number of organoids, similar in size and shape, are required for basic studies and translational applications. To address this requirement, researchers tried to develop liver organoids by coculture in suspension culture setups [13,14]. Those organoid-like spheroids were not functional enough, likely due to the lack of extracellular matrix (ECM) support to provide key cell–ECM interactions in cell-only aggregates. The ECM affects cell behavior and provides a proper niche and the biochemical cues required for the function of various cells in a specific organ [15]. Recent studies on decellularized liver tissue unveiled the role of liver ECM in activating hepatic differentiation and endodermal organoid formation [16,17,18,19]. Some studies have demonstrated that liver ECM promotes the maturation of human PSC-derived hepatocytes by downregulation of fetal liver markers (AFP and CYP3A7) and upregulation of other CYP genes as markers of metabolic activity [20,21,22]. Hence, an innovative strategy to establish liver organoids benefiting from both similar size and cell–ECM interactions is urgently required. In this regard, we previously developed a method for the encapsulation of three cell types in size-controlled alginate microcapsules enriched with sheep liver ECM. Although both coculture and ECM were effective on the functionality of hepatic cells, the migration and aggregation of the cells were restricted in the microcapsules, likely due to hindrances induced by the dense alginate matrix [20]. Here, we have developed an optimized, scalable 3D BLO culture system with a higher level of recapitulation of the liver-specific microenvironment by applying micropatterning technology. This was achieved by the addition of cell-sized MPs derived from liver ECM to the coculture of MSCs, HUVECs, and HE. We found that the expression levels of hepatic-specific genes ALB, G5PC, and CYP3A4 were significantly higher in BLOs compared to LOs. The maturation was further promoted upon intraperitoneal transplantation of BLOs in an acute liver injury mouse model, leading to an improved survival rate. Furthermore, efficient hepatic maturation was observed after ex ovo transplantation. Therefore, organoid engineering via the incorporation of cell-sized tissue-specific MPs within size-controlled multicellular aggregates provides a versatile strategy for the production of more functional organoids from different sources.

## 2. Materials and Methods

### 2.1. Ethical Statement

All experimental protocols using hESCs and hiPSCs primary cells (HUVEC and MSC), and animal studies were done in compliance with guidelines of the “Royan Institute Ethics Committee” (IR.ACECR.ROYAN.REC.1397.104).

### 2.2. Fabrication of Liver ECM-Derived MPs

Sheep-liver-derived ECM and its pre-gel solution were prepared according to a previously reported protocol, with a few modifications [23,24]. Briefly, after freezing then unfreezing sliced sheep liver was decellularized using 1% sodium dodecyl sulfate and 1% Triton X. Then, the decellularized tissue was lyophilized and milled. After that, the powder was enzymatically digested using 2% pepsin (Sigma, St. Louis, MO, USA) in 0.5 M acetic acid (Sigma) for 48 h at 4 °C to produce a solution of 20 mg/mL pre-gel of liver ECM.

The ECM-derived MPs were fabricated using a water-in-oil (*w/o*) emulsion method. Briefly, 250 mg of gelatin type B was dissolved in 5 mL of liver ECM solution (20 mg/mL) and stirred for 10 min at 40 °C to obtain a homogeneous solution of liver ECM–gelatin. Then, 2 mL of the resulting solution was added dropwise into 80 mL of corn oil and homogenized at 7000 rpm for 5 min to establish a *w/o* emulsion. Afterward, the mixture was chilled on ice for 1 h. For MP cross-linking, a glutaraldehyde solution (0.2% *w/w*) was added to the mixture and stirred at 700 rpm for 12 h. The resulting MPs were retrieved by centrifugation at 4000 rpm, treated with glycine solution (500 mM) to eliminate the residual aldehyde group, washed three times in deionized water, and used.

### 2.3. Characterization of Liver ECM-Derived MPs

The morphology of MPs was studied by scanning electron microscopy (SEM). The sample was dehydrated and gold-coated using a sputter coater (MSP-1S, Shinku Device, Ibaraki, Japan) and subsequently observed by SEM (SEM; VE-8800, Keyence, Tokyo, Japan). Furthermore, the size of MPs was measured manually for at least 150 MPs using phase-contrast micrographs and ImageJ software. To compare the chemical composition of the MPs with intact and decellularized liver tissues, Fourier-transform infrared spectroscopy (FTIR) (Ettlingen, Germany) was performed. All samples were lyophilized and milled, and FTIR spectra were obtained by a Bruker-Equinox 55 FTIR spectrometer equipped with attenuated total reflectance accessories.

To explore the degradation rate, MPs (10 mg, M1) were incubated at 37 °C in 10 mM Dulbecco’s phosphate-buffered saline (DPBS, 21600-010, Gibco, Waltham, MA, USA) (200 μL) at pH 7.4. Next, DPBS was discarded at a regular interval of 24 h, and the MP mass was measured (M2) [25]. Three samples of MPs were analyzed at each time point, and incubation was continued until the MPs were degraded (90% degradation was observed after 12 days). Finally, the percentage of degradation was calculated based on M2/M1 for each time point during the degradation process.

To determine possible cytotoxicity of MPs, human foreskin fibroblast viability was measured using an Orangu kit (OR01-500, Cell guidance systems, St. Louis, MO, USA) Fibroblasts were grown for 24 h under standard cell culture conditions in a 24-well plate at 5 × 10^4^ cells/mL initial concentration in DMEM medium (21331-020, Gibco) supplemented with 10% fetal bovine serum (FBS, 10270, Gibco), 0.1 mM nonessential amino acids (NEAA 35050-038, Gibco), 2 mM L-glutamine (35050-038, Gibco), and 1% penicillin and streptomycin (pen/strep, 15070-063, Gibco) [26,27]. The MPs were sterilized by UV exposure for 45 min, and the cells were cultured for an additional 48 h in medium containing different cell-to-MP ratios (1:1 and 1:2). The medium was then replaced with fresh medium containing Orangu, according to the manufacturer’s protocol. After 3–4 h of incubation at 37 °C, solution absorbance was measured at 495 nm.

### 2.4. Isolation and Expansion of Human-Umbilical-Cord-Derived Endothelial Cells

One donated human umbilical cord was washed with DPBS that contained 3% pen/strep. The umbilical vein was cannulated and washed with DPBS. Next, endothelial growth medium (EGM, CC-4147, Lonza) that contained 1 mg/mL collagenase IV (17104019, Gibco) was injected into the umbilical vein, followed by incubation at 37 °C for 15 min. The obtained cell suspension was sedimented at 1700 rpm for 10 min. The endothelial cells were resuspended in fresh EGM, transferred into a T25 flask, and cultured. The cells were used at passage number 1–3 [21]. All experiments were done in compliance with the guidelines of the Royan Institute Ethics Committee.

### 2.5. Expansion of Human Bone Marrow-Derived MSCs

The characterized cells were obtained from the Royan Stem Cell Bank (Iran) and cultured in low-glucose DMEM medium (11885-084, Gibco) supplemented with 10% FBS, 2 mM L-glutamine, 0.1 mM NEAA, 1% pen/strep, and 0.1 mM β-mercaptoethanol. The cells were used at passage number 1–4. All experiments were done in compliance with the guidelines of the Royan Institute Ethics Committee.

### 2.6. hPSCs Expansion and Differentiation towards Hepatic Endoderm in 3D Culture

Expansion and differentiation of human pluripotent stem cells (human embryonic stem cells (RH5) (RSCB0022) and human-induced stem cells (iPS4) (RSCB0082) were performed as previously described [28,29,30]. Briefly, 2.5 × 10^5^ cells/mL were transferred to a low attachment bacterial plate (628102, Greiner) with hPSC medium that was previously conditioned for 24 h on inactivated human foreskin fibroblasts with mitomycin C (M4287, Sigma-Aldrich). The medium contained DMEM/F12 (21331-020, Gibco), 20% knockout serum replacement (KOSR, 10828028, Gibco), 2 mM L-glutamine, 1% pen/strep, 0.1 mM NEAA, 1% insulin–transferrin–selenium (ITS, 41400045, Gibco), 0.1 mM β-mercaptoethanol (M7522, Sigma-Aldrich), and 100 ng/mL basic fibroblast growth factor (bFGF, GFH28-100, Cell GS). Only on the first day, 10 µM Rho-associated protein kinase (ROCK) inhibitor (Y0503, Sigma-Aldrich) was added to the dispersed cells. The plates were incubated under standard conditions (at 37 °C with 5% CO_2_ and saturated humidity), and the medium was replaced every other day.

The 4–5-day hPSC aggregates (average size 150 ± 20 μm) were induced to HE, following a published protocol [31]. Briefly, aggregates were rinsed with PBS plus Ca^2+^ and Mg^2+^ (PBS,14040117, Gibco) and cultured for 3 days in a definitive endoderm induction medium containing RPMI 1640 (5240041, Gibco), 0.1% bovine serum albumin (BSA, A9418, Sigma-Aldrich), and 1X B-27 without insulin (-Ins, 17504044, Gibco). The medium was supplemented with 6 μM CHIR99021 (CHIR, 04-0004-10, Stemgent, Beltsville, WV, USA) for one day. The aggregates were then washed with PBS and treated with 10 ng/mL Activin A (04-0004-10, R&D System) for two days. For HE differentiation, this medium was replaced with DMEM-F12, 2% KOSR, 10 ng/mL hepatocyte growth factor (HGF, 294-HG, R&D Systems, Minneapolis, MN, USA), and 10 ng/mL fibroblast growth factor 4 (FGF4, Royan Biotech, Tehran, Iran) for 4 days. All experiments were done in compliance with the guidelines of the Royan Institute Ethics Committee. 

### 2.7. Formation of Liver Organoids in Microwell Plates

Initially, hESC-derived HE aggregates were dissociated into single cells by 0.05 trypsin-EDTA (25300-062, Gibco). After that, different cell numbers per microwell (500 and 1000) at different cell/MP ratios (1:1, 2:1, and 3:1) were evaluated and set up, respectively (Appendix A). The MPs were mixed with HUVECs, MSCs, and HE at a 2:1 cell-to-MP ratio. After that, single cells at 10:7:2 ratios [27] of HE:HUVECs:MSCs were transferred to a microwell plate (AggreWell™, 34421, STEMCELL, Vancouver, Canada) containing approximately 7000 microwells per well (microwells were 400 μm in size). Before using the AggreWell™ plate, it was rinsed with an antiadherence rinsing solution (07010, STEMCELL) to ensure optimal performance. The cell suspension was centrifuged at 160 g for 5 min, and the resultant pellet was resuspended in a combination of hepatocyte complete medium (HCM, CC-4182, Lonza), DMEM/F12 as a maturation medium, and EGM medium (1:1 *v/v*) and then seeded on an AggreWell™ plate. The medium was supplemented with 2% KOSR, 10 ng/mL HGF, 10 ng/mL FGF4, 10 ng/mL oncostatin M (OSM, Royan Biotech), 0.1 mM dexamethasone (Dex, D-2915, Sigma-Aldrich, St. Louis, MO, USA), 1 mM NEAA, and L-glutamine and changed every other day. Next, the plate was centrifuged at 100 g for 3 min to capture the cells in the microwells. After 24 h, the condensed structures were harvested and treated for an additional 11 days in the same medium, which was changed every other day.

### 2.8. Incorporation of Liver-ECM-Derived MPs within Liver Organoids

To evaluate the rate of liver-ECM-derived MP incorporation within liver organoids and to visualize the distribution pattern of them in the organoids, a FITC-labeled MP suspension was added to the cell suspension prior to the pellet formation described above. The efficiency of incorporation was calculated by counting free MPs before and after aggregate formation using a hemocytometer. Furthermore, MP incorporation and distribution within the aggregates were traced by fluorescein isothiocyanate (FITC)-labeled MPs. After that, the aggregates were fixed, sectioned, and stained with 4′,6-diamidino-2-phenylindole (DAPI, Sigma-Aldrich, D8417), and micrographs were taken using a fluorescent microscope (IX7, Olympus, Tokyo, Japan).

### 2.9. RT-Quantitative Polymerase Chain Reaction (qRT-PCR)

Total RNA was isolated using an RNeasy Micro Kit (74004, QIAGEN, Hilden, Germany) according to the manufacturer’s instructions. Here, 1 μg RNA was used as a template for complementary DNA (cDNA) synthesis using a cDNA reverse transcription kit (4368813, Thermo Fisher Scientific, Waltham, MA, USA) according to the manufacturer’s instructions. For positive controls, we used the cDNA of fetal and adult livers. qRT-PCR was performed with sets of human-specific primers (Appendix A) and SYBR Green Master Mix (Takara Bio, Inc., SYBR Premix Ex Taq II RR081Q) using the StepOnePlus™ Real-Time PCR System. The results were normalized against glyceraldehyde 3-phosphate dehydrogenase (*GAPDH*) and calibrated against a combination of the same ratios of HE, HUVECs, and MSCs on Day 0 of the coculture. Data were analyzed and presented using the comparative CT method (2^−ΔΔCt^). All experiments were done in compliance with the guidelines of the Royan Institute Ethics Committee.

### 2.10. Tissue Processing and Immunofluorescence Staining

LOs and BLOs derived from hESCs were collected at the end of Day 12 and fixed by 4% paraformaldehyde (at 4 °C, overnight). The agar-embedded structures were processed and embedded in paraffin. After that, 6-μm sections were prepared and treated with antigen retrieval Dako (S2368, Glostrup, Denmark). The sections were incubated with primary antibodies in the blocking buffer overnight at 4 °C and then incubated with secondary antibodies at 37 °C for 1 h. The list of antibodies used in this work is shown in Appendix A. Finally, the nuclei were counterstained with DAPI. The micrographs were taken by a fluorescent microscope (IX71; Olympus) equipped with an Olympus DP72 digital camera.

### 2.11. Live & Dead Assay

Cell viability of the LOs and BLOs was evaluated using a LIVE & DEAD Viability Cytotoxicity Kit (Invitrogen, L3224, Carlsbad, CA, USA) on Day 14, according to instructions provided by the manufacturer. Briefly, the LOs and BLOs were washed with PBS and stained using calcein-AM (2 lM) and ethidium homodimers (4 lM) for 30 min at room temperature. LOs and BLOs were acquired using an inverted fluorescence microscope (Olympus, IX71).

### 2.12. Periodic Acid-Schiff (PAS) Staining

To evaluate the glycogen storage potential of LOs and BLOs derived from hESCs, periodic acid Schiff (PAS) staining was performed. Briefly, the LO and BLO sections were oxidized in 1% periodic acid for 5 min and then rinsed in dH_2_O. After that, the sections were treated with Schiff’s reagent for 15 min, followed by color development in dH_2_O for 5–10 min. Finally, micrographs were taken using a light microscope.

### 2.13. Indocyanine Green (ICG) Uptake and Release

For assessment of ICG uptake as a functional proof of hepatic organoid, both groups were incubated with 1 mg/mL ICG (12633, Sigma-Aldrich) for 1 h at 37 °C. Next, the LOs and BLOs derived from hESCs were washed three times with PBS and incubated in fresh medium for another 3 h for ICG release evaluation. The uptake and release of ICG were visualized using a light microscope [32].

### 2.14. Albumin (ALB) and Fibrinogen Secretion Assay

The ALB and fibrinogen secreted in the culture supernatants were respectively quantified using a human ALB-ELISA kit (Bethyl, E80-129) and a human fibrinogen-ELISA kit (Genway, 10-288-22856), according to the manufacturers’ instructions, by a microplate reader (Thermo Scientific, Multiskan Spectrum, 51118650). The results were normalized to the cell number.

### 2.15. Cytochrome P450 (CYP) Activity and Inducibility

To measure the activity of cytochrome P450 (CYP)-3A4, -2B6, -2C9, and -1A2 in both groups, lytic assays were performed by using a P450-GloTM assay kit (V8802, V8322, and V8792, respectively; Promega) based on the manufacturer’s instructions. The LOs and BLOs derived from hESCs were treated with basal media containing rifampicin (25 μM), valproate (100 μM), phenobarbital (100 μΜ), and omeprazole (100 µM) as CYP-3A4, -2B6, -2C9, and -1A2 inducers, respectively, or DMSO (0.1%) as the inducer solvent. The activity of each enzyme was measured by reading its luminescence using a luminometer (SpectraMax i3x), according to the manufacturer’s protocol.

### 2.16. Acute Toxicity Assay

The viability of cells in the LOs and BLOs derived from hESCs was evaluated by an Orangu kit after 48 h of exposure to four hepatotoxins, namely, paracetamol (APTP; 500, 1000, and 10,000 μM), fluorouracil (5-FU; 50, 100 and 500 μM), busulfan (10, 100 and 1000 μM) and tamoxifen (TAM; 1, 10 and 20 μM). For this purpose, culture media were refreshed every 24 h with serum-free media containing a certain concentration of each compound.

### 2.17. Ex Ovo Implantation of Bioengineered Liver Organoids on Chorioallantoic Membrane

The ex ovo implantation of BLOs derived from hESCs on CAM was performed based on a previously reported method [28]. The Fertilized Hy-line W-36 haying hens’ eggs were supplied by a commercial farm. The eggs were then cracked in a sterile laminar flow hood, and embryos with undamaged yolks were transferred into a surrogate shell that was bigger than the eggshell, sealed with plastic wrap, and incubated in a forced air incubator for 60 h at 37 °C with 60% humidity (Embryonic Day 0 (ED0)). On ED2.5, the yolk-embedded embryo was transferred to a second surrogate shell, which was bigger than the primary surrogate shell, and then sealed and incubated for another 5 days. One-day-old BLOs were implanted onto the surface of the CAM on ED7.5, distal from the embryo and proximal to major blood vessels, by gently scraping the upper CAM layer. Egg windows were sealed with plastic wrap, and the eggs were incubated for 14 additional days (ED 19.5). Finally, to study vasculogenesis, FITC-dextran (SLBT3224, Sigma-Aldrich) was injected into one CAM vessel, and other BLOs were subjected to qPCR and histology analysis. All experiments were done in compliance with the guidelines of the Royan Institute Ethics Committee.

### 2.18. Fabrication of a 3D-Printed Basket

To have a suitable container for the localization of the transplanted BLOs, a 3D basket device was fabricated using the fused deposition modeling (FDM) method, as described before, with minor modifications [29]. Briefly, the device was designed in CATIA V5 r19 and exported as an “STL” file. Then, it was sliced using Slic3r software. Based on the device dimensions and its vertical porosities, the key parameters were optimized for 3D printing. Printing temperature, nozzle diameter, printing speed, travel speed, and gap speed for this construct were set to 230 °C, 300 µm, 10 mm/min, 60 mm/min, and 60 mm/min, respectively. The porosity of the bottom of the device and the surface of the cap were set to 50% and 70%, respectively. The printing material was transparent polylactic acid (PLA, Befon). All 3D basket devices were printed by the Royan 3D printer (Royan Institute) (Appendix A).

### 2.19. In Vivo Implantation of Bioengineered Liver Organoids

We used the three following animal groups for the in vivo experiments: healthy mice (normal), acute liver injury model with transplanted empty basket (sham), and acute liver injury model with a transplanted basket containing BLOs derived from hESCs (treated). Acute liver injury in mice was induced by an intraperitoneal injection of 1 mL/kg (sublethal dose) carbon tetrachloride (CCl_4_) in 10-week old male C57BL/6 mice [30]. For immunosuppression, mice received a daily injection of cyclosporine (Novartis Pharmaceuticals, 20 mg/kg) for four days before transplantation until one week after transplantation. Next, two-week-old BLOs were transferred to the 3D-printed basket. The mice were then anesthetized by an intraperitoneal injection of 75 mg/kg ketamine (Alfasan) and 0.1 mg/kg medetomidine (Syva). After that, the 3D basket containing either approximately 1000 BLOs and that without BLOs (as BLO-transplanted or sham-transplanted, respectively), were transplanted into the abdominal cavity and sutured to the internal abdominal muscle. The animals were followed for 14 days for determination of survival rate. Furthermore, the sera levels of alanine transaminase (ALT) and aspartate transaminase (AST) on Day 2 after transplantation were evaluated using a commercially available kit (Biorexfars; BXC0215A and BXC0205A). For gene expression analysis, the samples were harvested from the 3D basket on Day 14. Moreover, human ALB in mice sera was quantified by a human ALB-ELISA kit (Bethyl; E80-129). All experiments were done in compliance with the guidelines of the Royan Institute Ethics Committee.

### 2.20. Chi-Square Analysis

This analysis was performed for the confirmation of survival rate significance. The hypothesis in this analysis was comprised of H0: the survival rate is not different between the two groups and Ha: the survival rate is different between the two groups. The decision rule was as follows: if Χ2 > 3.84, the null hypothesis is rejected (Appendix A).

### 2.21. Statistical Analyses

Data are presented as mean ± standard deviation (SD) from at least three biological replicates. Based on the normal distribution of data, statistical analyses were performed using the unpaired two-tailed Student’s *t*-test or one-way ANOVA to compare two or more groups, respectively. The assessments were followed by Tukey posthoc analysis to show the differences between groups. *P*-values less than 0.05 were considered significant. To evaluate the survival rate, a chi-square analysis was performed.

## 3. Results

### 3.1. Fabrication and Characterization of Liver-ECM-Derived MPs

In this study, we applied a *w/o* technique to generate liver-ECM-derived MPs (Figure 1A). The MPs were labeled with FITC for tracking after incorporation within the cell spheroids (Figure 1B). According to SEM findings, the MPs had a round and identical spheroid shape (Figure 1C). Size distribution analysis showed a narrow size distribution for the MPs, with a mean diameter of 7.9 ± 1.0 μm (Figure 1D). FTIR analysis was performed to compare the chemical composition of MPs and intact and decellularized liver tissues (Figure 1E). A comparison of the FTIR data of the MPs, decellularized liver ECM, and natural sheep liver confirmed the presence of collagen and proteoglycan in all of them. Within the region of 1500–1700 cm^−1^, three major peaks, including amide I (1652 cm^−1^), amide II (1542 cm^−1^), and amide III (1236 cm^−1^) bands, are associated with collagen protein [33]. The amide I band is mainly related to C=O stretching and N-H stretching, while the amide II band derives from N-H bending coupled with C-N stretching. The spectral range of 1000–1200 cm^−1^ mainly represents the C-O-C, C-O-H or C-C and C-C-O, and C-O-C stretches, vibrations of carbohydrate moieties from proteoglycan and polysaccharides, respectively [34]. To explore the stability of MPs under physiologic conditions, degradation over time was analyzed based on weight. The results demonstrated that the MPs were degraded over 11 days in a time-resolved manner (Figure 1F). To evaluate biocompatibility, fibroblasts were cultured on tissue culture plates in the presence of MPs at different cell/MP ratios. The results showed that following cell culture with different ratios of MPs, cell viability remained similar to that of the sham group (without MPs) (Figure 1G).

### 3.2. Differentiation of hPSCs towards Hepatic Endoderm

First, we optimized the differentiation protocol to generate HE from hESCs in terms of the induction period. Accordingly, the hESC aggregates were differentiated towards hepatoblasts in a 3D condition. Then, the aggregates were harvested on Days 3, 5, 7, and 9 of differentiation and evaluated. The qRT-PCR results demonstrated that the endodermal markers *SOX17*, *FOXA2,* and *CXCR4* were significantly downregulated (*p* < 0.05, Appendix A). However, on Days 7 and 9 of differentiation, the expression of HE-specific markers, including *HNF4α*, *HEX,* and *TBX3,* was significantly higher than that of Days 3 and 5 (*p* < 0.05, Appendix A). No significant difference was observed in the expression of HE markers between Days 7 and 9 of differentiation. Therefore, we applied a reverse screening experiment by comparing the hepatic markers’ gene expression on Days 7 and 9 for HE-derived liver organoids. We observed that the three mature hepatic genes, cytochrome p450 subunit 3A4 (*CYP3A4*), tyrosine aminotransferase (*TAT*)*,* and tryptophan 2,3-dioxygenase (*TDO*), were significantly expressed at higher levels in the LOs derived from Day-7 HE (*p* < 0.01, Appendix A). Therefore, the PSC-derived HE on Day 7 was selected for the coculture to form LOs and BLOs.

### 3.3. Formation of Liver Organoids in Microwell Plates

Initially, to optimize condensed aggregate formation, we examined various cell-to-MP ratios. Remarkably, cell condensation occurred only at 500 cell/microwell cell density with cell/MP ratio 2:1 (Appendix A).

Human ESC-derived HE on Day 7 was cocultured with BM-MSCs and HUVECs, with/without MPs, in AggreWell™ plates (Figure 2A). Twenty-four hours after coculture, the condensed aggregates were harvested and treated for further maturation for 12 days in static 3D culture (Figure 2A). Homogenous, round, and dense hepatic-like organoids were observed on Days 2, 7, and 12 post-coculture in the LOs and BLOs groups (Figure 2B). A gradual increase was demonstrated in size for both groups on Days 2, 7, and 12 post-coculture. The mean sizes on Days 2, 7, and 12 post-coculture were 150 ± 48.7, 250 ± 33.2, and 320 ± 45.3 µm, respectively. There was no difference in the mean size between the two groups (Appendix A). The results of MP incorporation indicated that the FITC-labeled MPs were successfully (approximately 80%) incorporated into the aggregates. Fluorescent images and microscopic sections showed that FITC-labeled MPs had a homogenous distribution pattern throughout the aggregates (Figure 2C).

### 3.4. Gene Expression Analysis of the Liver Organoids

To determine the gene expression profile in LOs and BLOs derived from hESC and hiPSC cultures after 12 days, we evaluated the mRNA expression of the following hepatic-specific genes: α-fetoprotein (*AFP*), albumin (*ALB*), tyrosine aminotransferase (*TAT*), tryptophan 2,3-dioxygenase (*TDO*), multidrug resistance-associated protein 2 (*MRP2*), multidrug resistance (*MDR*), asialoglycoprotein receptor1 (*ASGPR1*), carbamoyl phosphate synthase 1 (*CPS1*), glucose-6-phosphatase (*G6PC*) and phase I enzymes, cytochrome p450 subunit 1A2 (*CYP1A2*), subunit 3A4 (*CYP3A4*), subunit 3A7 (*CYP3A7*), subunit 2C9 (*CYP2C9*), and subunit 2B6 (*CYP2B6*). Moreover, we analyzed the gene expression of phase II and III metabolic enzymes, UDP-glucuronosyltransferase 2B15 (*UGT2B15*), UDP-glucuronosyltransferase-2B7 (*UGT2B7*), solute carrier organic anion transporter family member 2B1 (*SLCO2B*), organic anion transporting polypeptide 1B1 (*OATP1B1*), and aquaporin-7 (*AQP7*) (Figure 3). Interestingly, a significant downregulation of immature fetal hepatic markers, *AFP* and *CYP3A7*, was observed in the BLOs group compared to the LOs group (*p*< 0.01; Figure 3). Moreover, the expression of mature hepatic-specific genes (*ALB*, *MRP2*, *MDR*, *ASGPR1,* and *G6PC*) was significantly higher in the BLOs group than the LOs group. In addition, drug-metabolism-related genes, such as phase I enzymes (*CYP1A2*, *CYP3A4*, *CYP2C9,* and *CYP2B6*) as well as phase II and III enzymes (*UGT2B15*, *SLCO2B1*, *OATP1B1,* and *AQP7*), were significantly upregulated in the BLOs group compared to the LOs group (at least *p* < 0.05; Figure 3). We also examined the expression of *ITGa5* and *ITGb1* as genes of ECM receptors and their target genes (*FOXA2*, *HNF4α,* and *PXR*). The results indicated that their expression was significantly upregulated in the BLOs group compared to the LOs group (at least *p* < 0.05; Figure 3). Gene expression analysis was also performed for the liver organoids derived from HE that was differentiated from hiPSCs, and similar results were observed (Appendix A). These data suggest that engineering the microenvironment of liver organoids by the incorporation of liver ECM MPs could improve the maturation and metabolic capacity of the PSC-derived hepatocytes.

### 3.5. Immunostaining Analysis of the Liver Organoids

The result of live and dead staining indicated that most of the cells inside the LOs and BLOs were alive, and a small number of cells died (Appendix A). Immunofluorescence staining demonstrated that ALB, CYP3A4, CYP1A2, and E-CAD were expressed in both LOs and BLOs derived from the hESC groups. Z0-1 as an epithelial marker was also well-expressed in both groups. Moreover, vimentin and CD31-positive cells indicated the MSCs and HUVECs in both groups, respectively (Figure 4A). We also quantified and compared the protein expression in BLOs and LOs, respectively, for ALB (57 ± 5.6% vs. 34.3 ± 4.8%), CYP3A4 (51.6 ± 6.2% vs. 31.6 ± 5.1%), ALB/CYP3A4 (58 ± 5.2% vs. 34.3 ± 5.6%), CYP1A1 (33.3 ± 6.1% vs. 15.6 ± 5.3%), ZO-1 (46.6 ± 6.3% vs. 35 ± 5.2%), E-cadherin (44.3 ± 6.1% vs. 15.67 ± 4.5%), vimentin (9 ± 1.1% vs. 9.5 ± 1.01%), and CD31 (26 ± 1.5% vs. 26.5 ± 1.3%). The results indicated that there were more ALB+, CYP3A4+, CYP1A2+, ZO-1+, and E-cadherin+ cells in the BLOs group compared to the LOs group. Additionally, the percentage of vimentin+ cells and CD31+ cells in both groups was almost the same (Figure 4B). Primary human hepatocytes and adult fibroblasts were used as positive and negative controls, respectively (Appendix A).

### 3.6. Functional Analysis of the Liver Organoids

Functional analysis showed that hepatocyte-like cells in both groups (LOs and BLOs derived from hESCs) were able to uptake and release ICG (Figure 5A and Appendix A). PAS staining showed glycogen storage in both groups (Figure 5B). According to immunostaining results, the hepatic-like cells were localized in the marginal zone, and HUVECs and MSCs were localized in the center of the organoids, which was consistent with the PAS staining results. Moreover, ALB and fibrinogen secretion in the BLOs group was significantly higher than the LOs group (*p* < 0.05; Figure 5C,D). Moreover, our data showed that the activity of CYP3A4, CYP2B6, and CYP2C9 significantly increased (*p* < 0.05) upon treatment with drug inducers in both LOs and BLOs groups; however, CYP1A2 assessment did not show any significant difference (Figure 5E). Notably, higher drug inducibility was observed in the BLOs group compared to the LOs group (at least *p* < 0.05; Figure 5E). To evaluate their applicability in drug toxicity assessments, the BLOs and LOs groups were treated with different concentrations of various hepatotoxic compounds. The results showed that the susceptibility and sensitivity of the BLOs group towards the hepatotoxic compounds were significantly higher than the other group (at least *p* < 0.05; Figure 5F). Altogether, the BLOs group showed higher maturity and functionality than the LOs group; hence, we selected this group to perform subsequent experiments for further evaluations.

### 3.7. Further Maturation of Bioengineered Liver Organoids by Ex Ovo Transplantation

We next explored whether the highly vascularized microenvironment of CAM could improve the maturation of the BLOs derived from hESCs. To show the potential of vascularization and integration with the host vasculature, the chick was injected with dextran-FITC for visualization. The comparison of fluorescence microscopy images, taken before and after dextran-FITC injection, showed multiple blood vessels in the close vicinity of the BLOs after 14 days (Appendix A). Moreover, H&E staining showed the penetration of capillaries into the implanted BLOs (Appendix A). Gene expression analyses demonstrated that at 14 days after transplantation, early hepatic development genes were significantly downregulated while late hepatic genes were significantly upregulated, and the BLOs were closer to the adult liver in terms of gene expression patterns (Appendix A). Thus, the transplantation of BLOs into a highly vascularized microenvironment could promote their maturation. Therefore, CAM could enhance the maturation of BLOs by providing a highly vascularized microenvironment.

### 3.8. Ectopic Transplantation of Bioengineered Liver Organoids in Mice with Acute Liver Injury

Acute liver injury and transplantation procedures are illustrated in Figure 6A. BLOs derived from hESCs were transplanted in mice in a 3D-printed basket. The baskets were highly porous in order to facilitate mass transfer and engraftment between the transplanted organoids and the host (Figure 6B). In order to investigate the capacity of BLOs to improve survival rates in the mouse model, the baskets with/without BLOs were transplanted into the internal abdominal muscles of cyclosporine-treated mice with acute liver injury. Animal survival rate was evaluated for 14 days post-transplantation; the results demonstrated that in the BLO-transplanted group, all animals survived, while 60% in the sham group were alive (Figure 6C). The result of chi-square analysis indicated that Χ2 = 5.48, which exceeds the critical value of 3.84. Therefore, we reject H0. We have significant evidence, α = 0.05, to show that the two survival curves are different.

In the next step, we measured changes in liver injury indicators, ALT and AST, two days after transplantation. Data showed significant decreases in the plasma levels of ALT (Figure 6D) and AST (Figure 6E) in mice transplanted with baskets containing BLOs compared to the sham group transplanted with empty baskets. Moreover, to explore BLO functionality, we measured human ALB in mice sera. Using an ELISA test, 20–35 ng/mL human ALB was detected in the sera of mice transplanted with baskets containing BLOs (Figure 6F). Relative gene expression analysis showed that two immature assigned genes (*AFP* and *CYP3A7*) were significantly decreased compared to the in vitro cultured, two-week-old BLOs (at least *p* < 0.05). In contrast, the expression of many mature genes was significantly increased to near that of the adult liver compared to the in vitro cultured, two-week-old BLOs (Figure 6G). All primers used in this section were human-specific.

## 4. Discussion

Bioengineered organoids have broad applications in experimental medical research, and they are progressively opening up new avenues in drug screening, toxicology, and disease modeling. Nowadays, only 10% of new components that enter phase I clinical trials proceed to the next phases. This emphasizes the need to develop and optimize methodologies that reduce the expenses, time, and preclinical animal studies needed to identify a suitable compound and perform drug screening [35,36]. Recently, advances in technologies involving the production of organoids from human PSCs have presented a great opportunity for developing a more reliable, rapid, and cost-effective drug-screening platform compared to the currently used animal and in vitro models [37,38,39].

Liver development is a complex process in which distinct microenvironmental and biophysical signals are crucial. Human PSC technology has promoted in vitro recreation of processes that occur during liver organogenesis [35]. Takebe et al. took advantage of this approach and generated liver organoids using a coculture system and a Matrigel matrix, a solubilized basement membrane preparation extracted from Engelbreth–Holm–Swarm mouse sarcoma cells [12]. However, current organoid fabrication technologies have faced some drawbacks, such as the nonspecific matrix provided by Matrigel, uncontrolled size, and poor reproduction [36]. Bioengineering approaches using liver-specific ECM and micropatterning systems have been proposed as novel strategies to overcome these limitations [37]. Herein, we addressed this challenge by incorporation of cell-sized sheep liver ECM-derived MPs within cell aggregates during aggregate formation in microwell plates. The MPs were fabricated by chemical cross-linking of a water-in-oil dispersion of digested decellularized liver tissue. Then, HE cells were cocultured with MSCs and HUVECs in AggreWell™ plates, with or without the MPs, to form 3D structures named bioengineered liver organoids (BLOs) and LOs, respectively. Furthermore, advances in the micropatterning of liver organoids can potentially open an avenue to screen thousands of compounds/drugs to narrow down potential candidates. This model is more suitable for in vitro drug screening and disease modeling rather than cell-based therapy.

In this study, we successfully developed a simple and efficient method to produce liver-ECM-derived MPs with a round shape and homogenous size in the range of the typical size of single cells (~8 µm). Moreover, FTIR assessment of the LEM gel, MPs, and liver ECM verified the preservation of the key components of liver tissue after processing into the liver ECM solution and hydrogel MPs, which was consistent with previously reported data [1,23].

Gene expression analysis showed significant upregulation of specific hepatic genes (*ALB*, *TAT*, *MRP2*, *TDO*, *MDR*, *ASGPR1*, *CPS1,* and *G6PC*) as well as phase I (*CYP1A2*, *CYP3A4*, *CYP2C9,* and *CYP2B6*), phase II, and phase III enzyme genes (*UGT2B15*, *UGT2B7*, *SLCO2B*, *OATP1B1,* and *AQP7*) in the BLOs compared to the LOs group. In parallel, the hepatic immature genes (*AFP* and *CYP3A7*) in the BLOs group were downregulated compared to the LOs group. These results agree with other reports that have demonstrated that liver ECM successfully improves hepatocyte differentiation. In this context, mouse liver scaffolds could enhance the differentiation of hiPSC-derived hepatocytes [16]. Huanjing and colleagues showed that porcine liver ECM promoted the hepatic differentiation of BM-MSC [38]. Moreover, our results showed that hPSC-derived hepatocytes in the BLOs group expressed epithelial membrane protein markers like E-cadherin and ZO–1, implying that compared to hepatocytes in the LOs group, they were more polarized. This observation is consistent with previous studies, which have shown that liver ECM and supportive cells enhance the epithelialization and polarity of hepatocytes and other epithelial cells [40,41,42,43]. Moreover, based on our findings, the BLOs group was more functionally mature than the LOs group, as shown by higher glycogen storage, ability to uptake and release ICG, and CYP induction. We also found significant upregulations of ALB and fibrinogen secretion (as two important factors secreted by mature hepatocytes) in the BLOs group compared to the LOs group. This was comparable to ALB secretion by hepatospheres in the study of Pettinato and colleagues (450 vs. 230 ng/mL, respectively) [44]. Moreover, ICG uptake by hepatocytes happens through organic anion-transporting polypeptides (OATP1B1 and OATP1B3), which are expressed on the sinusoidal membrane of hepatocytes and ICG and then excreted into bile canaliculi via multidrug resistance associated protein 2 (MRP2). The successful ICG uptake and release indicated that ICG absorption and clearance occurred after 1 and 3 h, respectively. The results were consistent with the gene expression of transporters. In contrast to previous studies, ICG release took a shorter time (i.e., 3 h versus approximately 6 h in other studies) [45,46]. Previous studies have shown that recapitulation of the native microenvironment can improve the maturation and functionality of different cells [47,48,49]. Our finding shows that BLOs are more sensitive in the evaluation of drug-induced cytotoxicity compared to LOs. These data suggest that the BLOs could detect the toxicity of the reactive metabolites generated by drug-metabolizing enzymes such as CYP enzymes. This is important because, in many cases, drug-induced hepatotoxicity is caused by the reactive metabolites produced by drug-metabolizing enzymes [42]. Likewise, ex ovo and in vivo ectopic transplantations of the BLOs resulted in their further maturation and functionality. We found that the application of chick CAM provided a highly vascularized microenvironment. Ectopic transplantation also enhanced the BLOs’ maturation, as indicated by the markedly increased expression of hepatic maturation genes. We assumed that the vascular structures that surrounded the BLOs through the highly porous 3D basket enhanced the maturation of the BLOs. Some studies have demonstrated the effect of in vivo and ex ovo vascularizations on the maturation of organoids or other tissues [33,34,50,51,52]. Here, we found that the transplanted BLOs could rapidly perform hepatic functions such as ALB secretion; in addition, ALT and AST levels in BLO-transplanted mice were significantly decreased compared to the sham group. These findings demonstrate that the BLO-transplanted group successfully performed metabolic detoxification and biosynthesis of albumin and other crucial biomolecules in acute liver injury and eliminated toxic metabolites to provide the healthy microenvironment required for liver regeneration, resulting in an improved survival rate. The ability of BLOs to quickly perform hepatic functions may be attributed to the presence of liver-ECM-derived MPs and the fact that the BLOs were surrounded by vessels.

Liver-specific ECM components provide a natural microenvironment that promotes and mimics cell–cell and cell–matrix interactions similar to the liver [45]. Cells interact with ECM via integrins, which are cell adhesion receptors that regulate cellular behavior [53]. The ECM provides guiding cues for better phenotype maintenance, differentiation, proliferation, and polarization of hepatocytes both in vitro and in vivo [47]. Studies have demonstrated that ECM hydrogels derived from different organs, such as heart [48,49], liver [54], skeletal muscles, and kidney [51], could serve as a proper cell culture platform to improve cell functionality and maturity. In our previous studies, liver ECM was used as a hydrogel [21] and patch [22] to improve the functionality of hepatic cells. The reconstruction of an ECM in a 3D culture of microscale cell spheroids is a challenging subject. The collagen-cell-sized MPs can be incorporated into the inner space of the cell spheroids and work as microenvironment regulators due to their physicochemical properties [55].

There is evidence that integrins have other functions besides acting as cell adhesion molecules, and it is likely that interactions between specific receptors and distinct ligand binding sites can convey unique information to the cells [56]. Guo and colleagues demonstrated that the endothelial cell matrix could upregulate the expression of *FOXA2*, *HNF4α,* and *PXR* and, thus, modulate the expression of several downstream functional targets, including CYP450 enzymes and transporters, promoting the functional and metabolic maturation of hepatic-like cells [57]. Therefore, in the present attempt to understand the liver ECM–hepatocytes cross-talk and the molecular mechanism through which it could promote the functionality and maturation of BLOs, we examined the expression of integrin α5 (*ITGa5*) and β1 (*ITGb1*), the genes of the ECM receptors in the cells, and their target genes (*FOXA2*, *HNF4α,* and *PXR*). We found that their expression significantly increased in the BLOs group compared to the LOs group. Therefore, we assumed that the effects of the liver-ECM-derived MPs on the functionality and maturation of the BLOs could be a result of α5β1 integrin and natural biomolecule upregulation in MP-mediated signaling pathways. 

Despite the promising results, the present work still has to consider and address additional issues in future studies, with the aim to consolidate the relevance of the data presented here. The size of the organoids should be controlled to avoid impairing the quality of cells that are located in the core. Hence, in vitro culture conditions should be improved to limit the merging of LOs and BLOs that we noticed after two weeks in 3D static-suspension conditions. An advanced analysis of the molecular interactions between MPs and cells is mandatory to appraise the beneficial experimental conditions on both the morphology and physiology of the cells. Additional investigations into the bioengineered liver organoids matured subsequently to ex ovo transplantation should help in getting deep information on the microstructure and maturation aspects of the organoids as well as the quality of anastomosis. Finally, for in vivo transplantation studies, the engraftment level of those organoids should be evaluated in order to appraise if the effects noticed on the markers of the liver injury induced in the mouse model are related to the implanted differentiated cells or their paracrine activity.

To the best of our knowledge, this is the first study reporting the generation of cell-sized organ-specific MPs derived from the native ECM and their application in engineering an organoid microenvironment. We tried to recapitulate the original niche for HE by using MPs derived from liver ECM and a 3D coculture system to generate more functional and mature hepatocyte-like cells in a BLO manner. These liver-ECM-derived MPs also provide a scalable platform for the mass production of BLOs. Therefore, organoid engineering via the incorporation of cell-sized tissue-specific MPs within size-controlled multicellular aggregates provides a versatile strategy for the production of more functional organoids from different sources for biomedical applications.

## Figures and Tables

**Figure 1 cells-10-01274-f001:**
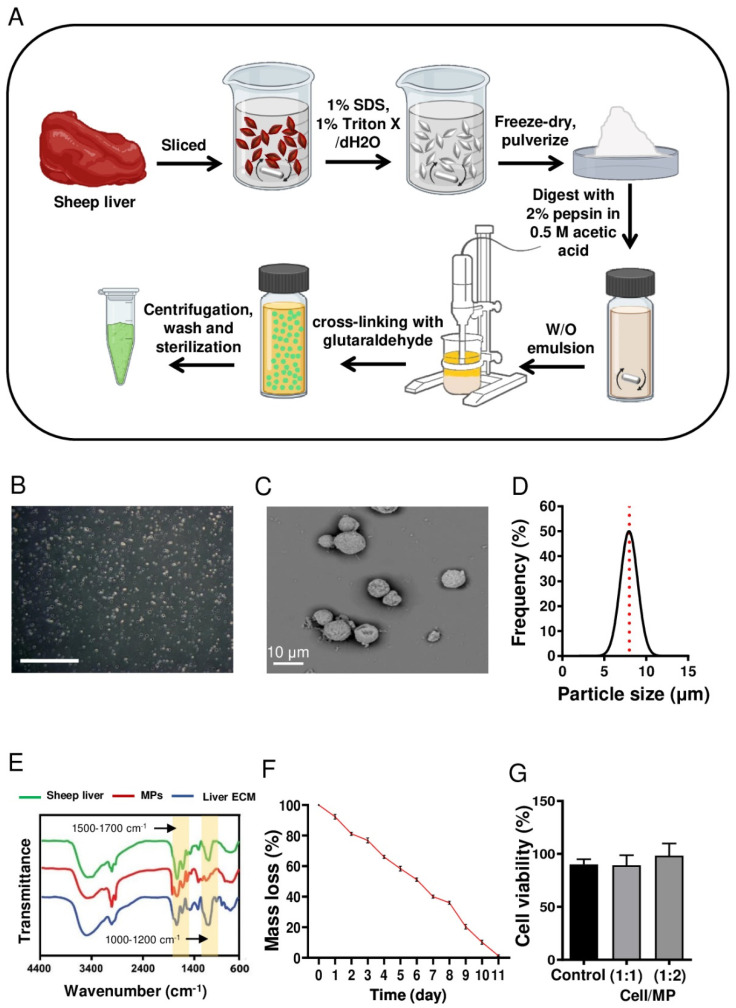
Fabrication and characterization of liver-ECM-derived microparticles (MPs). (**A**) A schematic presentation of liver-ECM-derived MP fabrication. (**B**) Phase-contrast microscopy image of MPs (scale bar: 500 μm). (**C**) A representative scanning electron microscopy (SEM) image of the MPs. (**D**) Size distribution histogram of the MPs. (**E**) Fourier-transform infrared spectroscopy (FTIR) spectra of intact and decellularized liver tissues and liver-ECM-derived MPs. (**F**) Mass loss of MPs in DPBS (pH 7.4, at 37 °C). (**G**) Cytocompatibility analysis of MPs by MTS assay on fibroblast cells cultured for 48 h on tissue culture polystyrene, with/without MPs, at different cell/MP ratios. (ECM: extracellular matrix; MPs: microparticles; SEM: scanning electron microscopy; FTIR: Fourier-transform infrared spectroscopy; DPBS: Dulbecco’s phosphate-buffered saline).

**Figure 2 cells-10-01274-f002:**
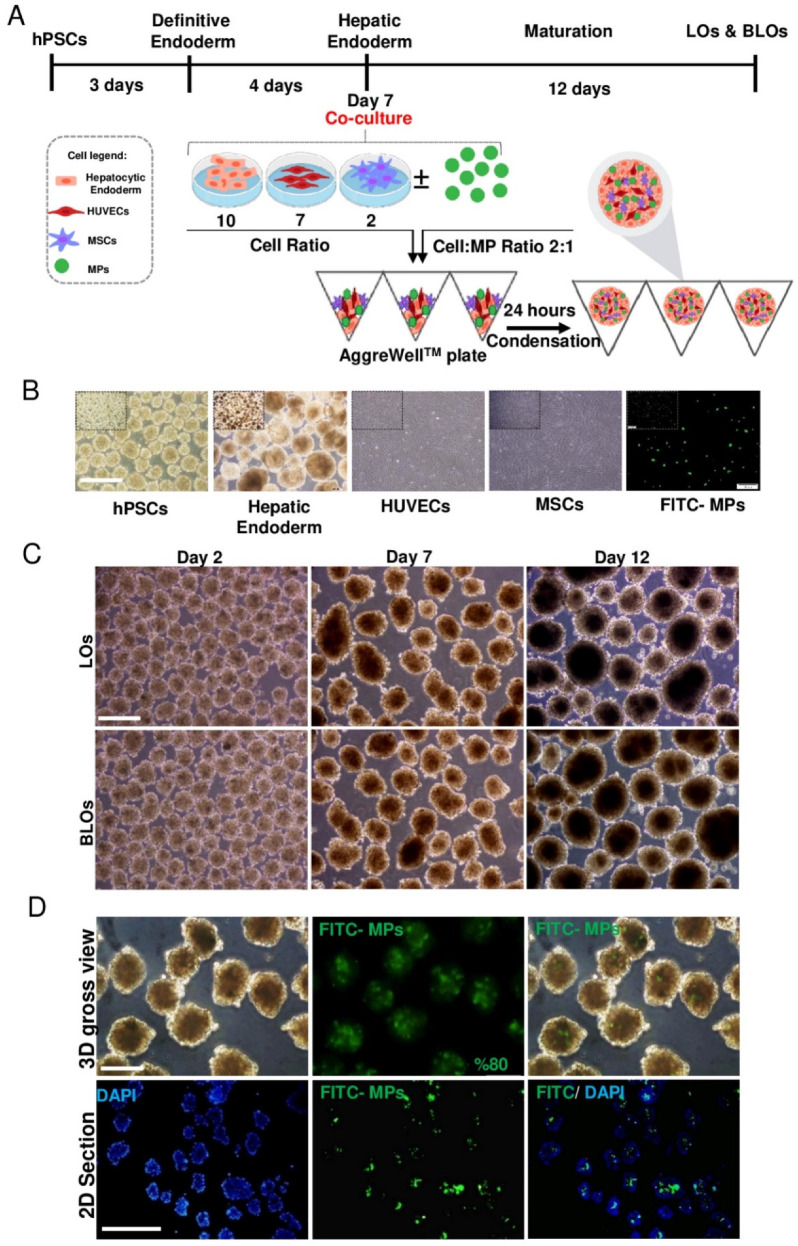
Formation of LOs and BLOs from human pluripotent stem-cell-derived hepatic endoderm. (**A**) A schematic presentation of the stepwise protocol for differentiation of hESCs (RH5 cell line) towards HE and their coculture with HUVECs and MSCs (10:2:7 of HE:MSC:HUVEC), with/without MPs (2:1 cell:MP), in AggreWell™ plates. (**B**) Phase-contrast microscopy images of hPSCs, HE derived from hESCs, HUVECs, MSCs, and FITC-MPs (scale bar: 200 µm) (**C**) Phase-contrast microscopy images of cell aggregates on Days 2, 7, and 12 post-coculture (scale bar: 200 μm). (**D**) Visual illustration of MP incorporation within cell aggregates composed of three types of cells (HE, HUVECs, and MSCs; 10:7:2, respectively) using fluorescein isothiocyanate (FITC)-labeled MPs. Phase-contrast microscopy images (top left), fluorescence microscopy images (top middle), merged fluorescence microscopy image (top right), and paraffin-embedded cross-section (bottom) of the MP-incorporated cell aggregate. FITC-labeled MPs are visualized in green. The nuclei of cells were counterstained with DAPI. Scale bar: 200 (top) and 500 μm (below). HE: hepatic endoderm; HUVECs: human umbilical vein endothelial cells; MSCs: mesenchymal stem cells; LOs: liver organoids; BLOs: bioengineered liver organoids; hESCs: human embryonic stem cells; MPs: microparticles; FITC: fluorescein isothiocyanate.

**Figure 3 cells-10-01274-f003:**
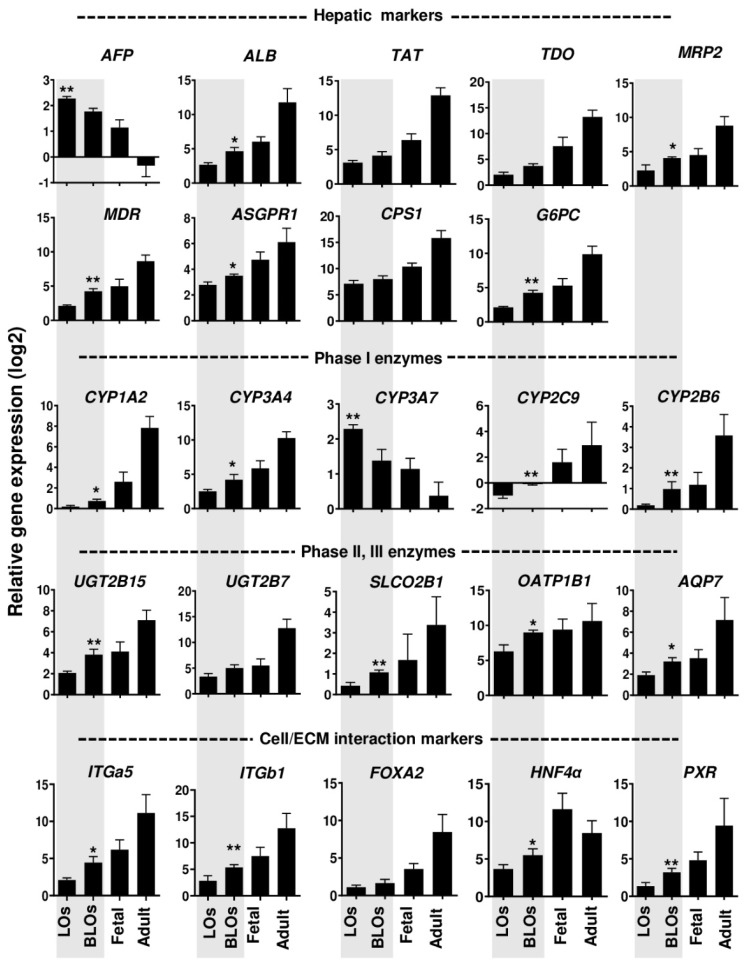
Gene expression of hepatic signatures in the LOs and BLOs derived from hESC-HE. Relative gene expressions of the hepatic-specific genes and the genes related to phases I, II, and III of drug metabolism in the BLOs and LOs groups compared to fetal and adult human livers as control groups (*n* = 4). Data were normalized against *GAPDH* and are presented as fold changes calibrated to expression values on Day 0. Data are shown as mean ± SD (*n* = 3). Statistical analysis was performed using unpaired two-tailed Student’s *t*-tests. * *p* < 0.05 and ** *p* < 0.01. LOs: liver organoids; BLOs: bioengineered liver organoids; hESC: human embryonic stem cell (RH5 cell line); HE: hepatic endoderm.

**Figure 4 cells-10-01274-f004:**
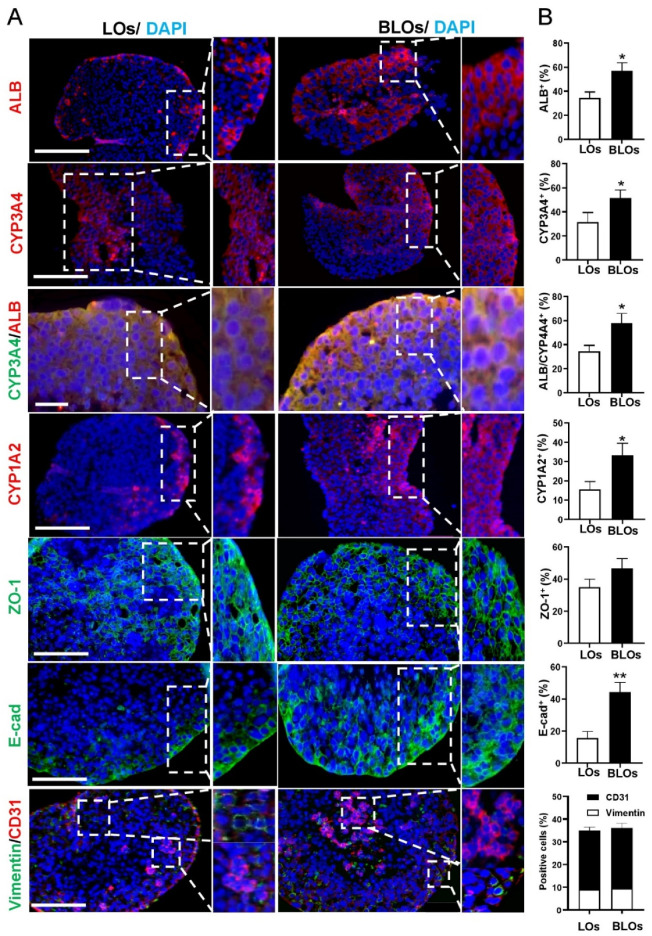
Protein expression profile of the LOs and BLOs groups. (**A**) Immunostaining of BLOs and LOs derived from hESCs for AFP, ALB, CYP3A4, CYP1A2, ZO-1, E-cadherin, and vimentin/CD31. The nuclei were counterstained with DAPI (scale bar: 200 and 100 μm). (**B**) The ratios of ALB^+^, CYP3A4^+^, ALB/CYP3A4^+^, CYP1A2^+^, ZO-1^+^, E-cadherin^+^, vimentin^+^, and CD31^+^ cells in all DAPI-stained cells obtained from at least 4 views per section for at least 3 sections * *p* < 0.05 and ** *p* < 0.01. LOs: liver organoids; BLOs: bioengineered liver organoids.

**Figure 5 cells-10-01274-f005:**
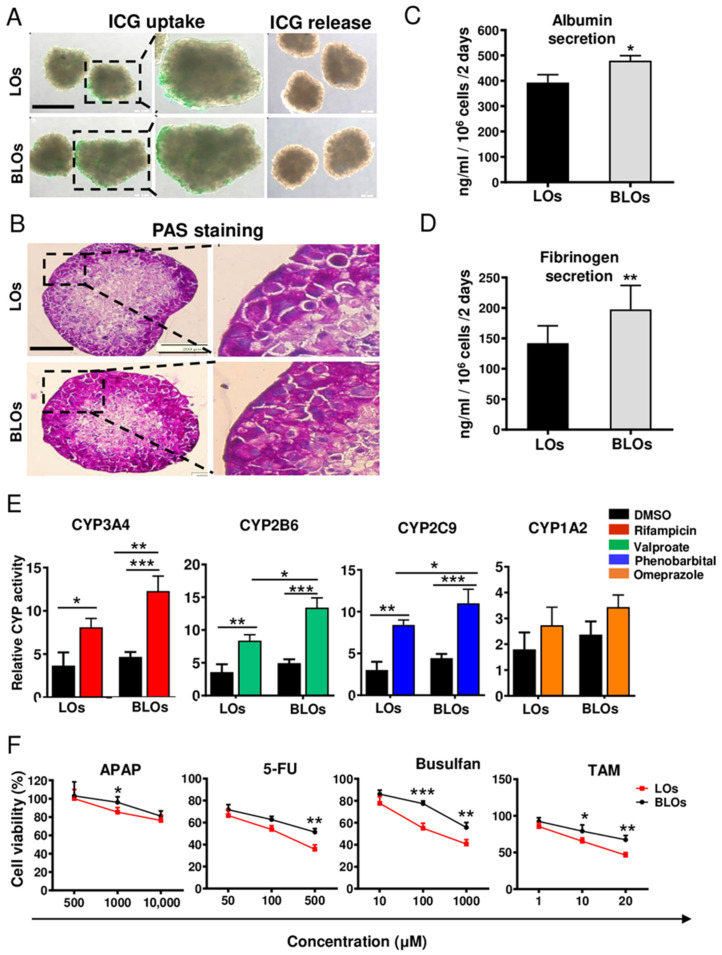
Functional assays of the LOs and BLOs derived from the hESC groups. (**A**) The BLOs and LOs groups were examined for their ability to take up and release indocyanine green (ICG). The ICG uptake and release took 1 and 3 h, respectively. Scale bar: 200 μm. (**B**) Periodic acid Schiff (PAS) staining was done to analyze glycogen storage. Scale bar: 200 μm. (**C**,**D**) Quantification of albumin (ALB) and fibrinogen secretion after 48 h. Data are shown as mean ± SD (*n* = 3). Statistical analysis was performed by using unpaired two-tailed Student’s *t*-tests. * *p* < 0.05 and ** *p* < 0.01. (**E**) Induction of CYP3A4, CYP2B6, CYP2C9, and CYP1A2 by rifampicin, valproate, phenobarbital, and omeprazole, respectively, after 72 h of treatment. (**F**) Cell viability in the BLOs and LOs groups was assessed by Orangu^®^ kit after 48 h of exposure to different concentrations of hepatotoxic compounds, paracetamol (ATAP), fluorouracil (5-FU), busulfan, and tamoxifen (TAM). Data are shown as mean ± SD (*n* = 3). Statistical analysis was performed using one-way ANOVA, followed by Tukey posthoc. * *p* < 0.05, ** *p* < 0.01, and *** *p*< 0.001. LOs: liver organoids; BLOs: bioengineered liver organoids.

**Figure 6 cells-10-01274-f006:**
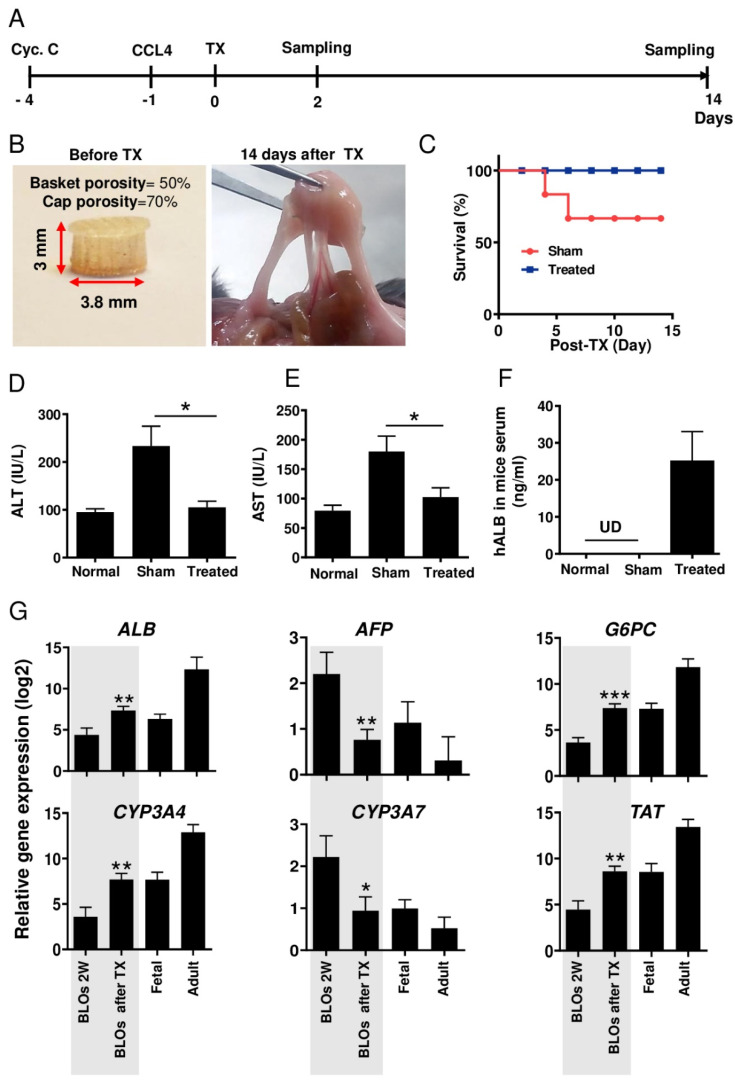
Ectopic implantation of the BLOs in mice with acute liver injury. (**A**) The timeline representing the time points in the figure. (**B**) 3D-printed basket containing the BLOs, before (left) and after (right) transplantation. (**C**) A graphic presentation of the survival rate of the control and treated groups. (**D**,**E**) Biochemical assessment of blood sera from mice with acute liver injury treated with the BLOs. BLO transplantation led to significantly decreased alanine transaminase (ALT) and aspartate transaminase (AST) levels. (**F**) Human ALB was measured by ELISA in mice sera. Normal: normal mice; Control: CCl_4_-treated mice transplanted with empty baskets; Treated: CCl_4_-treated mice transplanted with 3D baskets containing approximately 1000 BLOs (*n* =8). Data are shown as mean ± SD (*n* = 3). Statistical analysis was performed by using unpaired two-tailed Student’s *t*-tests. * *p* < 0.05. (**G**) Relative gene expression of hepatic-specific genes (*ALB*, *AFP*, *G6PC*, *CYP3A4*, *CYP3A7*, and *TAD*) in the BLOs, before and after transplantation, and fetal and adult liver tissues as control groups. Data were normalized against *GAPDH* and are presented as fold changes compared with data obtained on coculture Day 0 as the calibrator. Data are shown as mean ± SD (*n* = 8). Statistical analysis was performed by using unpaired two-tailed Student’s *t*-tests. * *p* < 0.05, ** *p* < 0.01, and *** *p*< 0.001. BLOs: bioengineered liver organoids; 3D: three-dimensional; ALT: alanine transaminase; AST: aspartate transaminase; ALB: albumin; CCL4: carbon tetrachloride.

## Data Availability

All data supporting the findings of this study are available within thearticle and its Appendix A or from the corresponding author upon reasonable request.

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
