# Peer review of "Tissue-Specific Microparticles Improve Organoid Microenvironment for Efficient Maturation of Pluripotent Stem-Cell-Derived Hepatocytes"

_cells, 2021, doi:10.3390/cells10061274_

Round 1

Reviewer 1 Report

Cells 1163560

Tissue-Specific Microparticles Improve Organoid Microenvironment for Efficient Maturation of Pluripotent Stem Cell-derived Hepatocytes

Zahmatkesh et al

Abstract and introduction excellent

Major:

My major comments are positive. The authors described the hurdle for upscaling (lines 644-645), but have done important. Test to evaluate the functionality of the novel cell-sacffold contructs.

Minor:

M&M2.9, would have been better if the authors stick to MIQE=precise guidelines (more genes to normalize and not only GAPDH which has an unstable expression) especially since relative expression differences are sometimes significant but less than 2-fold. It will only tke one or two additional qRTPCR to have a more robust relative expression.

M&M.10 What is the negative control in IF?

M&M2.19 Did you correct for multiple measurements? I must admit statistics is noy mny cup-of-coffee

Results 3.1 (also for M&M2.2)

How was it checked that the Triton treated sheep liver was completely decellularized?

Results 3.4 Expression in adult liver is sometimes much higher than LO or BLO, is that of a concern?

Results 3.6 and figure 5, why is a comparison with liver or freshly isolated primary hepatocytes missing?

Line 565 human

Discussion:

Would it be possible (or maybe have even more impressive results) if decell and pulverized material and the different cell types are all from the same species?

Reviewer 2 Report

Zahmatkesh et al. have generated PSC-derived liver organoids using tissue-specific microparticles. This paper is informative, but much improvement is needed. Please address the following comments.

  1. In figure 3, fetal and adult liver should be analyzed with n = 3. There are large individual differences in the expression of drug-metabolizing enzymes.
  2. Figure 5E: CYP1A2 mRNA can be induced by omeprazole treatment?
  3. Does CYP have metabolic activity? I think that CYP evaluation at the activity level is also essential.
  4. Are the primers used in figure 6G human-specific?
  5. Immunostained images of LO and BLO after transplantation should be shown.

Reviewer 3 Report

In the present manuscript, Zahmatkesh and colleagues developed an organoid system based on co-culture of human PSC-derived hepatocytes, mesenchymal cells and HUVECs, with the addition of liver-derived microparticles providing ECM components. Organoids were tested in vitro and ex ovo; moreover, transplantation within acute liver injury mouse models, via a 3D printed carrier, assessed the functional capabilities of transplanted organoids in the setting of liver injury. The Authors developed an interesting platform for in vitro disease modelling and drug screening analysis, and the efficiency of their system in replicating the functional capabilities of hepatocytes was also suggested by the results in the mouse model of liver injury. However, there is limited translatability of the Authors’ findings in human cell therapy, mainly due to the use of PSC-derived hepatocytes. Other methodological aspects require attention by the Authors.

1) The Authors developed their BLOs by differentiating PSC-derived hepatic endoderm cells into mature hepatocytes in vitro. However, the translation of this approach into human clinics is problematic, both for embryo-derived human PSC (due to ethical reasons) and for induced PSCs (since iPSCs undergo extensive manipulation and can therefore harbor tumorigenic risk). This aspect should be acknowledged as a limitation of their study, and Authors should not refer to their organoid system as candidate option for human cell therapy.

2) Authors should include RRID codes for cells obtained from biobanks or for cell lines. In particular, Authors should clarify which cells were used for each set of experiment. In Methods, Authors refer to RH5 and iPS4 cells, but in results, hESCs are mentioned as well. This should all be clarified and properly referenced in Methods.

3) The Authors observe that organoids embedded with MPs are characterized by an improved maturation and metabolic profile, and identify a modification in integrin profile as the possible mediating mechanism for this effect. However, Authors should discuss which MP component(s) might be responsible for this interaction, as well as performing double immunofluorescence stains in order to identify which cell population within the organoids is expressing the integrins after MP exposure. This is important in order to assess whether MPs have a direct effect on HEs, or whether their effects are actually mediated by modulation of supporting cells (i.e. HUVECs and mesenchymal cells) within organoids.

4) Authors performed their experiments on cell cultures in the form of organoids in suspension by culturing their cells as 3D aggregates. However, organoids appear very large in size, which can have implications in the penetration of compounds and nutrients within the whole structure. To validate their findings, the following aspects should be addressed by the Authors: i) Organoid size can affect functional capability of the organoid, since only the outer layers of cells would be implied in drug metabolism in vitro; in this light, ICG images in figure 5A are hard to interpret, as the outer cells appear to have a strong green staining, but this cannot be properly assessed for the cells within the center of the organoids. ii) Cell viability appears to be compromised by the excessive dimensions of the organoids; this can be seen in PAS stains, where the cells within the core of the organoids appear extensively damaged and necrotic, more than “PAS-negative”. iii) Although the Authors seed the cells at a fixed ratio, organoid expansion might be driven by one cell population over the others, which can affect the variations in cell phenotype observed by the Authors more than cell differentiation itself. In light of these comments, the Authors should perform live-dead assays on organoids at different culture time points, and identify the most appropriate organoid size to achieve high cell viability. Moreover, relative evaluation of each cell population should be assessed also in the light of the following comment.

5) For a large body of results, the Authors rely on gene expression analysis of organoids to assess the differentiation of HEs into mature hepatocytes. However, relative gene expression can be influenced by the presence of other cell types in cultures. Therefore, the Authors should also characterize the variations in HUVEC and mesenchymal cell markers in their culture conditions, in order to confirm that increased mature hepatocyte marker expression is not due to a reduced presence of other cells (i.e. HUVECs and mesenchymal cells) within organoids.

6) The Authors individuate a different ICG uptake in BLOs compared to LOs. Methods for ICG uptake quantification and actual data should be included in the manuscript. Moreover, in light of the previous comments, Authors should assess whether there is any correlation between organoid size and ICG uptake, further supporting their evaluation of the best organoid size for their scopes.

7) The Authors should evaluate differentiation markers in Figure 3 by immunohistochemistry and immunofluorescence as well. Particularly, expression of canalicular markers such as MDR would help visualize whether cell polarization is occurring, as suggested by the Authors in the Discussion. Moreover, the double IF stain for ALB and CYP3A4 in figure 4 is not very convincing. Besides the double stain, Authors should include separate channels to better show the expression of single markers in the same cells.

8) The Authors investigated the capability of their BLOs to rescue liver injury in a murine model of acute liver damage. However, the characterization of cell engraftment in this model needs to be improved regarding the following aspects. i) Do grafted cells migrate out of the basket? Is the liver colonized or are cells ectopically located within the abdominal cavity? Tracing techniques should be employed to investigate grafted cell fate. In general, visualization of cells within graft is of pivotal importance, and immunophenotypical analysis of cells from the graft (with markers of cell fate and of proliferation or vitality) should be performed as well. ii) The vascularization that the Authors observed macroscopically should be evaluated by histology – IHC stains for vessel markers. iii) Although mice are treated with cyclosporine, is there any immune reaction to the graft which could affect cell microenvironment?

9) The evaluation of the effects of BLO transplantation in mice should be better characterized by the Authors. In particular, the Authors observe an improvement in ALT/AST profile in transplanted vs sham mice, but no other assessment of liver injury is performed. Moreover, it is unclear how BLOs transplanted within the 3D-printed basket would improve liver injury. Authors should assess liver histology, in order to evaluate liver pathological modifications in sham mice and their eventual reversal in BLO-transplanted mice, also investigating (as in previous comment) the possibility of transplanted cells engraftment within mice liver. Furthermore, other liver functionality tests should be performed, in order to fully characterize the effects of transplanted cells in the mouse model.

10) Authors report different mortality rates in sham vs BLO-treated mice. Authors should provide the number of mice in each experimental group, and statistical significance of the differences observed in mortality by chi-squared analysis. Moreover, Authors should assess whether any significant difference in mortality can be observed between sham mice and CCl4 mice which did not undergo any surgical procedure, in order to assess a possible role of the procedure on mice outcomes.

11) The H&E stains in Supplementary Figure 7 are not convincing, as arrows seem to point to cell aggregates more than capillaries as stated by the Authors. Immunohistochemistry for blood vessel markers should be included, as well as BLO cell markers (both hepatocyte and supporting cells) to visualize transplanted cells within the CAM.

12) For the quantification of hepatocyte marker expression (Figure 3/6), the Authors used fetal and adult human livers. This should be included in Methods, particularly including tissue source and related ethical statement. Moreover, primary hepatocytes and hepatoblasts seem more appropriate controls than whole livers.

13) An informed consent statement should be included for all used human tissues and cells.

Minor comments:

  • Result section 3.2: The Authors should include microscopy imaging of hESCs-HE aggregates.
  • In figure 1B, Authors should include high magnification of immunofluorescences of FITC-labelled microparticles.
  • Page 3-4 (lines 117-129): this part should be moved to the beginning of the Discussion and replaced with a brief summary of the Aims of the study.
  • Supplementary figure 4 is not referred in the main text.
  • In figure 4, the panels with magnified area appear to be distorted; please correct this in the revised manuscript.
  • Panel labelling within Supplementary Figure 7 legend should be corrected.

Round 2

Reviewer 2 Report

The authors have partially modified the manuscript, but they did not address to most of the comments pointed out at the initial review. Please address the comments below.

  1. In figure 3, fetal and adult liver should be analyzed with n = 3. There are large individual differences in the expression of drug-metabolizing enzymes. Error bars are necessary for fetal and adult liver. Total RNA samples of fetal and adult liver are commercially available. I think there is no ethical concerns related to purchasing these samples.
  2. Figure 5E: Can CYP1A2 be induced by omeprazole treatment? Please perform additional experiments.
  3. Please describe that the primers used in figure 6G are human specific in the materials and methods section.

Author Response

Reviewer 2

The authors have partially modified the manuscript, but they did not address to most of the comments pointed out at the initial review. Please address the comments below.

1-In figure 3, fetal and adult liver should be analysed with n = 3. There are large individual differences in the expression of drug-metabolizing enzymes. Error bars are necessary for fetal and adult liver. Total RNA samples of fetal and adult liver are commercially available. I think there is no ethical concerns related to purchasing these samples.

Answer: Thank you for your comment. To address this concern, more samples of human adult and fetal liver were used (n=4). The error bars were included and the results updated in the revised version of the manuscript (Fig.3) ( and Supp.Fig.5).

2-Figure 5E: Can CYP1A2 be induced by omeprazole treatment? Please perform additional experiments.

Answer: Thank you for your suggestion. The inducibility for CYP1A2 using omeprazole was evaluated. As compared to other CYPs, no significant induction was measured. The results are included in figure 5E in the revised version of the manuscript.

3-Please describe that the primers used in figure 6G are human specific in the materials and methods section. 

Answer: The information “All primers used in this section were human specificis now highlighted in the revised version of the manuscript.

Reviewer 3 Report

In their point-by-point reply, the Authors addressed the Reviewer’s comments mostly by discussing their experimental setting, but very few modifications have been made to the main body of the manuscript. Overall, the Reviewer’s impressions on the manuscript remain unchanged, and the Authors should take more time to carefully revise their manuscript and to address the Reviewer’s comments.

Some aspects of the manuscript were discussed by the Authors in the point-by-point reply; however, besides replying to the Reviewer, these observations and novel data (for example the ones from comment 10 of the review) should be included in the main text of the revised manuscript as well.

This latter aspect is particularly important for the live-dead assay performed by the Authors, which is of pivotal importance and must be included in the revised manuscript. From a technical point of view, the core positivity is not convincing, and a nuclear contrast should be added when capturing the images as it is very surprising that not a single dead cell is visible in the images. High magnifications should be provided. Most importantly, the Authors used the exact same image for LOs and BLOs, raising serious concerns about the validity of these data.

Author Response

Reviewer 3

Comments and Suggestions for Authors

In their point-by-point reply, the Authors addressed the Reviewer’s comments mostly by discussing their experimental setting, but very few modifications have been made to the main body of the manuscript. Overall, the Reviewer’s impressions on the manuscript remain unchanged, and the Authors should take more time to carefully revise their manuscript and to address the Reviewer’s comments.

Answer: Thank you for your valuable feedback. We have taken into consideration all comments raised.

1- Some aspects of the manuscript were discussed by the Authors in the point-by-point reply; however, besides replying to the Reviewer, these observations and novel data (for example the ones from comment 10 of the review) should be included in the main text of the revised manuscript as well.

Answer: Thank you for your comment. The updated information is included in the main text of the revised version of the manuscript [materials and methods (lines 369-373) as well as in results’ section (lines 571-573)]. Also, the method of calculation and tables were inserted in the supplementary materials’ section (supplementary tables 4, 5, and 6).

2- This latter aspect is particularly important for the live-dead assay performed by the Authors, which is of pivotal importance and must be included in the revised manuscript. From a technical point of view, the core positivity is not convincing, and a nuclear contrast should be added when capturing the images as it is very surprising that not a single dead cell is visible in the images. High magnifications should be provided. Most importantly, the Authors used the exact same image for LOs and BLOs, raising serious concerns about the validity of these data.

Answer: Thank you for your attention and we sincerely apologize for the mistake. The related photos were revised and updated (Supplementary figure 4).

Round 3

Reviewer 2 Report

This paper can be accepted in the current version. Excellent work.

Author Response

We thank the reviewer #2 for the constructive comments and advices that helped us to improve the quality of our ms

Reviewer 3 Report

In the revised version of the manuscript, the Authors addressed the main major concerns raised in the revision.   However, the authors should include in the Discussion a few comments regarding the aspects of their Manuscript that were not addressed by their revision and remained unchanged compared to the first version of their work. In the reviewer's opinion, these issues remain of pivotal importance and the fact they were not completely addressed represents a limitation in the Authors' work, which should be acknowledged in the Manuscript and taken into account when evaluating the impact and relevance of the presented results.    Specific aspects to be addressed and included in the Discussion as limitations in the revised manuscript are: 1. Excessive size of organoids impairs the quality of cells in the core. 2. MP elements and interactions with organoid cells have not been investigated in detail. 3. Results on ex ovo model are preliminary and should be confirmed by immunostains. 4. No characterization of cell engraftment in the murine model was performed, no information is provided on liver injury in transplanted and in sham mice, and no mechanism is proposed for the beneficial effects on liver function in transplanted mice.

Author Response

We thank the reviewer #3 for the rigorous review and for the constructive comments raised that helped us to improve the quality of our ms.

To address the final comments, we have inserted the paragraph here below in the discussion section (Lines 704-717):

Despite of the promising results, the present work still has to consider and address additional issues for the future studies with the aim to consolidate the relevance of the data presented here. The size of the organoids should be controlled to avoid impairing the quality of the cells that located in the core. Hence, in vitro culture conditions should be improved to limit the merging of the LOs and BLOs that we have noticed after two weeks in 3D static suspension conditions. An advanced analysis of the molecular interactions between MPs and cells is mandatory to appraise the beneficial experimental conditions on both morphology and physiology of the cells.

Additional investigations on the bioengineered liver organoids matured subsequently to ex ovo transplantation should help in getting deep information on the microstructure and maturation aspects of the organoids a well as on the quality of anastomosis. Finally, for in vivo transplantation studies, engraftment level of those organoids should be evaluated in order to appraise if the effects noticed on the markers of the liver injury induced in the mouse model are related to implanted differentiated cells or on their paracrine activity.